# Prescription Practices and Usage of Antimicrobials in a Tertiary Teaching Hospital in Rwanda: A Call for Antimicrobial Stewardship

**DOI:** 10.3390/antibiotics13111032

**Published:** 2024-10-31

**Authors:** Acsa Igizeneza, Leopold Bitunguhari, Florence Masaisa, Innocent Hahirwa, Lorette D. Uwamahoro, Osee Sebatunzi, Nathalie Umugwaneza, Ines Pauwels, Ann Versporten, Erika Vlieghe, Ayman Ahmed, Jean Claude S. Ngabonziza, Caroline Theunissen

**Affiliations:** 1Department of Microbiology and Parasitology, University of Rwanda, Huye P.O. Box 117, Rwanda; 2Department of Clinical Biology, University of Rwanda, Kigali P.O. Box 3286, Rwanda; drbitunga@gmail.com (L.B.); masaisa2009@gmail.com (F.M.); ulorette@gmail.com (L.D.U.); sebatunzi12@gmail.com (O.S.); cldsemuto@gmail.com (J.C.S.N.); 3Department of Internal Medicine, University Teaching Hospital of Kigali, Kigali P.O. Box 655, Rwanda; 4Department of Pharmacology and Toxicology, University of Rwanda, Kigali P.O. Box 655, Rwanda; innocenthahirwa@gmail.com; 5Department of Pharmacy, University Teaching Hospital of Kigali, Kigali P.O. Box 655, Rwanda; 6Department of Accident and Emergency, University Teaching Hospital of Kigali, Kigali P.O. Box 655, Rwanda; 7Department of Surgery, University Teaching Hospital of Kigali, Kigali P.O. Box 655, Rwanda; umugwanezanathalie@gmail.com; 8Laboratory of Medical Microbiology, Vaccine & Infectious Disease Institute (VAXINFECTIO), Faculty of Medicine and Health Sciences, University of Antwerp, 2610 Antwerp, Belgium; ines.pauwels@uantwerpen.be (I.P.); ann.versporten@uantwerpen.be (A.V.); erika.vlieghe@uza.be (E.V.); 9Department of General Internal Medicine, Infectious and Tropical Diseases, Antwerp University Hospital, 2650 Edegem, Belgium; 10Faculty of Medicine and Health Sciences, University of Antwerp, 2000 Antwerp, Belgium; 11Unit of Applied Medical Sciences, Faculty of Medical Laboratory Sciences, University of Khartoum, Khartoum 11111, Sudan; ayman.ame.ahmed@gmail.com; 12Rwanda Biomedical Centre, Kigali P.O. Box 7162, Rwanda; 13Research, Innovation and Data Science Division, Rwanda Biomedical Centre, Kigali P.O. Box 7162, Rwanda; 14Department of Biomedical Sciences, Institute of Tropical Medicine, 2000 Antwerp, Belgium; 15Department of Clinical Sciences, Institute of Tropical Medicine, 2000 Antwerp, Belgium; ctheunissen@itg.be

**Keywords:** antimicrobial stewardship, low- and middle-income countries, point prevalence survey, Rwanda tertiary hospital, teaching hospital, antimicrobial resistance

## Abstract

**Background:** Antimicrobial resistance (AMR) is a global problem that results in high morbidity and mortality, particularly in low- and middle-income countries. Inappropriate use of antimicrobials is a major driver of AMR. This study aimed to evaluate the rate and quality of antimicrobial prescription and use at the University Teaching Hospital of Kigali (CHUK), a tertiary-referral teaching hospital. **Methodology:** A point prevalence survey (PPS) of antimicrobial prescription was conducted using the Global PPS tool, including a healthcare-acquired infection (HAI) module. **Results:** On the day of the PPS, 39.3% (145/369) of inpatients were prescribed at least one antimicrobial. Out of the 259 prescribed antimicrobials, 232 (89.6%) were antibacterials, of which 151 (65.1%) belonged to the watch group of the WHO AWaRe classification. The top three antibiotics prescribed were cefotaxime (87; 37.5%), parenteral metronidazole (31; 13.4%), and meropenem (23; 9.9%). Stop or review dates for the prescribed antimicrobials were documented in 27/259 prescriptions (10.4%). Surgical prophylaxis (SP) was prescribed for longer than one day in 83.3% of 61 patients. Samples for culture were sent for 27.1% (63/232) of all the patients prescribed antibiotics. **Conclusion:** This PPS demonstrates multiple indicators of the poor use of antimicrobials, including the high prevalence usage of watch antibiotics and prolonged surgical prophylaxis and other poor-quality indicators. Thus, there is an urgent need for intervention to improve antimicrobial stewardship.

## 1. Introduction

Antimicrobial resistance (AMR) is an increasing global health challenge, ranked fifth among ten WHO-highlighted threats affecting public health [1]. AMR caused an estimated 1.27 million deaths globally in 2019, and this is projected to increase to 10 million deaths annually by 2050 in the absence of effective action to change the current trends towards increasing resistance [2,3]. In 2022, an alarming report, compiled from the 76 countries in the Global Antimicrobial Resistance and Use Surveillance System (GLASS), demonstrated that, on average, 42% of *E. coli* isolates are third-generation cephalosporin-resistant, and 35% of *Staphylococcus aureus* isolates are methicillin-resistant *“*https://www.who.int/publications/i/item/9789240062702 (accessed on 2 September 2024)”.

The existing evidence demonstrates clearly that inappropriate prescription and use of antimicrobials are major drivers of the development and spread of AMR [4]. In 2015, the WHO set a global action plan for AMR, with antimicrobial stewardship (AMS) in human health, defined as the promotion of the rational usage of antimicrobials, identified as one of the key strategies within the plan [5]. 

Healthcare providers may contribute to the emergence of AMR in multiple ways. These include irrational antimicrobial prescribing in both hospital and community settings, insufficient infection prevention and control practices within healthcare facilities, and admitting immunosuppressed patients prone to multi-drug-resistant organism (MDRO) infections to hospital inappropriately [6,7,8,9,10,11,12].

In addition, low-and-middle-income countries (LMICs), including Rwanda, face additional challenges. These include shortages of highly skilled personnel, particularly those trained in infection prevention and control, and the limited awareness among other healthcare providers about best practices in the prescription and use of antibiotics. Furthermore, the high prevalence of presentations due to infectious diseases in these settings, together with the limited hygiene and sanitation, reduced microbiological laboratory capacity, and limitations in the availability and quality of medicinal products contribute to the significant burden and threat posed by AMR to LMIC [13]. 

Despite the growing evidence of an AMR threat at the national level in Rwanda, there are still sparse recent data available at the hospital level in the country [14,15,16,17]. A recent study conducted in three referral hospitals demonstrated a resistance rate above 70% of *Klebsiella* and *Acinetobacter* ssp. isolated from bloodstream infection against all tested antibiotics [18]. In a study conducted by Bizimungu et al. at CHUK, it was demonstrated that *Enterobacterales* species were resistant to ceftriaxone in 72% and ciprofloxacin in 66% of cases, and *Acinetobacter* ssp. were resistant to imipenem in 80% of cases [19]. In Rwanda, AMS programs have been initiated at several hospitals to try to optimize the usage of antimicrobials and reduce antimicrobial resistance, but the programs are not yet fully implemented. 

At the time of this study, no data on antimicrobial use in Rwandan hospital settings, particularly at tertiary and teaching hospitals like CHUK, were available. Therefore, this study aimed to furnish baseline information on the prescription and usage of antimicrobials at the hospital level to guide the development and implementation of evidence-based antimicrobial stewardship programs for Rwanda.

## 2. Results

### 2.1. The Prevalence of Antimicrobial Usage at Hospital and Departmental Levels

A total of 369 inpatients were enrolled in this study, of which 282 (76.5%) were adults, 75 (20.3%) were children, and 12 (3.2%) were neonates. Sixty-one patients (16.5%, *n* = 369) had been referred from other healthcare centers, and 41 (67.2%) of these were already on antimicrobials before their current hospitalization at CHUK. The hospital-wide prevalence of patients on antimicrobials was 39.3% (145/369), and 97% (141/145) of the patients on antibiotics received at least one antibiotic. A total of 81 patients (58.9%) received more than one antimicrobial. Figure 1 illustrates the antimicrobial prevalence stratified by the healthcare units in which patients were hospitalized including medical, surgical, and intensive care units for adults, children, and neonates (Figure 1). High antimicrobial prevalences were noted in patients in the pediatric surgical ward (PSW), pediatric intensive care unit (PICU), and neonatology intensive care unit (NICU) with prevalences of 68.8% (11/16), 80% (8/10), and 75% (9/12), respectively. The lowest antimicrobial prevalence was observed in the adult medical ward (AMW) with 28.3% (41/145), with intermediate figures of 40.5% (51/126) in the adult surgical ward (ASW) and 54.5% (6/11) in the adult intensive care unit (AICU). The remaining pediatric wards had prevalences of 30% (3/10) in the hematology–oncology ward (HOPMW) and 41% (16/39) in the medical ward (PMW).

Overall, the most prescribed antibiotic classes were beta-lactams (147/232, 63.4%), followed by imidazole derivates (31/232, 13.4%,) and glycopeptides (GP) (17/232, 7.3%,). Sixty-four percent (95/147) of the prescribed beta-lactams consisted of third generation cephalosporins (3GCS), mainly cefotaxime (*n* = 87), and ceftriaxone (*n* = 8), followed by carbapenems (23/147, 15.6%), mostly meropenem. The distribution of the five most prescribed antibiotics at ward level is illustrated in Figure 2. Carbapenem prevalence was 22.2% (4/18) in the PSW, 37.5% (3/8) in the AICU, and 23.5% (4/17) in the PICU. In the PICU, glycopeptide prevalence was 41.2% (7/17).

Antibiotics within the WHO access group represented 35% (81/232); conversely, those within the watch group represented 65% of all prescriptions. The ICU departments had the highest prescription rate for use of watch antibiotics 72% (31/43), followed by the surgical wards 64.3% (63/98) and the medical wards 59% (56/95). Sixty-four percent (96/151) of all prescribed watch antibiotics were 3GCS, with 57.6% (87/151) of them being cefotaxime, followed by carbapenems (meropenem) with 15.2% (23/151) and glycopeptides (vancomycin) with 11.2% (17/151). The most prescribed antibiotic from the access group was metronidazole (IV) at 43.2% (35/81) (Figure 2).

### 2.2. Indications for Antimicrobial Usage

In total, 178 out of 259 (68.7%) antimicrobials were prescribed for the treatment of infections, whereas 81 out of 259 (31.3%) were for prophylaxis, of which 61 (75.3%) of these were for surgical prophylaxis. The top five indications for antimicrobial treatment were respiratory infections (RI), including bronchitis and pneumonia (20.2%, 36/178), followed by skin and soft tissue infections (SSTI) (14.6%, 26/178), tuberculosis (10.7%, 19/178), intra-abdominal infections (IA) (9%, 16/178), and central nervous system infections (CNS) and sepsis with 7.3% each (13/178). Figure 3 illustrates the distribution of antibacterials prescribed for the top five indications for treatment. The top four indications for prophylaxis were for medical and surgical gastro-intestinal conditions (30.8%, 25/81), bone and joint (18.5%, 15/81), and medical prophylaxis for maternal risk factors (MP-MAT) and newborn risk factors (NEO-MP) such as very low birth weight (13.6%, 11/81 each).

Sixty-one (61) antimicrobials were prescribed to 42 patients for surgical prophylaxis. The most prescribed antimicrobials were 3GCS (62.3%,38/61) and metronidazole (13.2%, 8/61). Only 1.6% (1/42) of the patients received a first generation cephalosporin (1GC). Of all 42 patients for whom surgical prophylaxis was prescribed, 7 (16.6%) received a single dose, while the remaining 35 (83.3%) received multiple doses extending to more than one day of prophylaxis.

Out of 178 prescribed antimicrobials for treatment, 71% (127) were for community-acquired infections (CAI), while 29% (51) were for hospital-acquired infections (HAI). The overall prevalence of HAI in the hospital was 9.5% (35/369 patients). However, among patients receiving antibiotics, this prevalence of HAI increased to 24.8% (35/141 patients). The main HAI were surgical site infections (SSI) (37.1%, 13/35), followed by non-intervention-related HAI other than BSI (31.4%), intervention-related infections (20%), and bloodstream infections (BSI) (11.4%). The top three antibacterials prescribed for CAI were third generation cephalosporins (3GCs) (38.8%, 40/103), metronidazole (16.5%, 17/103), and glycopeptides (10.7%, 11/103). For hospital-acquired infections (HAI), the most frequently prescribed antibacterials were 3GCs (28.6%, 14/49), followed by carbapenems (26.5%, 13/49) and metronidazole (12.2%, 6/49).

Two-hundred and four out of 259 (78.8%) prescribed antimicrobials were administered parenterally compared to 54 (20.8%) and 1 (0.4%) orally and intramuscularly, respectively. The parenteral use of antimicrobials exceeded 90% in the pediatric department and 70% in the adult department. The reason for prescription was present in the patient file for the treatment of 239 (92.3%), but only 105 (40.5%) prescriptions were based on an elevated biomarker of infection (WBC, CRP). For 62 (24%) prescriptions, a body fluid or tissue sample was sent to the microbiology lab for culture and antimicrobial sensitivity testing (AST).

In the subset of patients receiving targeted antimicrobial treatment, the commonly isolated bacteria were *Escherichia coli* 12/22 (54.5%), among which 41.7% were extended-spectrum beta-lactamases producing bacteria (ESBL). Additionally, 8.3% were third-generation-cephalosporin-resistant bacteria (3GCREB), and 33.3% were both, ESBL and 3GCREB. They were followed by *Klebsiella pneumoniae* and *Acinetobacter* spp. with 13.6% (*n* = 3) each, and their phenotypic resistances were 33.3% ESBL and 33.3% carbapenem-resistant non-fermenters (CR-NF), respectively. Only two (9%) *Staphylococcus aureus* were isolated, and neither showed resistance. Other isolated bacteria included one *Enterococcus* spp. and one *Pseudomonas aeruginosa*, with a resistance profile of vancomycin-resistant enterococci (VRE) and CR-NF, respectively. Antimicrobials were prescribed for empiric use in 220 (85%) cases, and only 34 (15.5%) of these prescriptions had sent samples to the microbiology laboratory for culture and AST. The patients missed one or more doses for 17.4% (45/259) of prescribed antimicrobials. Guidelines were only able to be assessed for cases treated for tuberculosis; other prescriptions of antimicrobial were made without reference to any guidelines, as no local guidelines exist. Table 1 includes the data of the different quality indicators of antimicrobial prescription in the pediatric and adult departments.

## 3. Discussion

This PPS of antimicrobial use is the first to be conducted in a Rwandan hospital and gives valuable baseline information on antimicrobial use at a tertiary care facility in a LMIC settings. Antimicrobial prescriptions at CHUK for most indications cannot be based upon any guidelines, as no guidelines exist; however, the study revealed multiple markers of poor antibiotic prescribing practices, including the widespread use of third generation cephalosporins and meropenem, frequent multi-day use of antibiotics for prophylaxis, and very infrequent use of planned time-limited therapy. Often antibiotics were prescribed empirically upon arrival for critically ill patients with suspected sepsis, which may be appropriate, but appropriate microbiological samples to guide therapy were frequently not obtained first. As CHUK is a tertiary referral hospital, some referred patients had also received empirical antibiotic therapy prior to admission.

This PPS identified several areas which could be the subject of AMS improvement efforts at CHUK. The overall prevalence of antimicrobial use (38%) is comparable to observations in other tertiary hospitals from the East African region such as a tertiary-care referral hospital in Tanzania (38% [20]), but it is lower than observed prevalences in certain hospitals in other African countries (62.3%; 67.7% [21,22]). However, there was a wide variation in the prevalence between the different healthcare units within the hospital, suggesting major differences in clinical practice. The prevalence of antimicrobial use in patients reached 55 to 80% in some units, including the adult ICU, pediatric ICU, and pediatric surgical ward. This variation could be attributed to several factors, including a higher rate of bacterial infections among specific population groups, problems with increased prevalence of healthcare-acquired infections (such as intravascular device-related and surgical site infections) in these healthcare units, and the differing implementation of effective AMS and infection prevention and control measures between the different healthcare units.

Sixty-five percent (65%) of all the prescribed antimicrobials belonged to the watch group of the WHO AWaRe classification. This rate is comparable to that seen in the Democratic Republic of Congo (67.1%, [23]). However, compared to LMICs in other regions, this is relatively high [24,25,26,27]. This may be due to different factors such as the limited knowledge of the WHO AWaRe classification and AWaRe antibiotic guidelines and the broad antibacterial spectrum of watch antibiotics. Others contributing factors might include the limited availability of alternative access antibiotics, the lack of local treatment guidelines, and difficulties with microbiological diagnosis, as well as the lack of a stewardship program in SSA [28,29,30,31]. Within the study’s ICU settings, the presence of many severely ill patients with serious infections requiring long hospital stays, possibly with multi-resistant organisms, may have contributed to high watch prescription usage [32].

As in many other WHO regions, 3GCS (e.g., ceftriaxone and cefotaxime) were the most prescribed antimicrobials, followed by metronidazole [22,33,34]. This was the case for treatment, as well as for prophylaxis of infection. This finding is a concern given the role of 3GCS in the selection of extended-spectrum beta-lactamase (ESBL)-producing bacteria [35,36]. The highest rate of 3GCS prescription was recorded in adult surgical wards (55.3%), with prevalences comparable to reports from countries such as Kenya (55%) and Bangladesh (44.6%) [22,33]. A key pattern driving their high usage is their perceived role in surgical prophylaxis, despite international guidelines recommending first generation cephalosporins (with or without metronidazole, depending on the type of surgery), which can adequately cover most bacteria involved in surgical site infections [37]. The high prescription level of 3GCs might also be due to their use in the treatment of postoperative infections, because of their broad-spectrum activity in both Gram-positive and Gram-negative bacteria [38].

Aside from the high level of 3GCS prescription for surgical prophylaxis, the duration of surgical prophylaxis exceeded one day in 84% of the patients. This finding is lower than findings from Tanzania, Uganda, and Zambia that reported prolonged SP in more than 96% of the cases [39]. International guidelines as well as the WHO antibiotic guidelines recommend, for most cases, a single dose of surgical antibiotic prophylaxis with cefazolin for surgical interventions not exceeding 4 h [37,40]. The prolonged SP at CHUK could be explained by different factors such as the prescription behavior of healthcare providers, fear of post operation SSI, inadequate infection prevention control (IPC) measures, the lack of a local SP guideline, little awareness of the WHO SP guidelines, and poor recording of stop and review dates.

In addition, 3GCS and other watch antibiotics were prescribed for community-acquired infections in 56% of the cases, whereas the WHO antibiotic guide and AWaRe classification often recommends otherwise [41]. This prescription behavior might be explained by the lack of local guidelines, inadequate microbiological diagnostics, and poor AMR surveillance, which otherwise might be available to inform adequate, locally adapted, and empiric antimicrobial prescription.

Eighty-five percent of all antimicrobials were prescribed empirically. A possible explanation lies in the fact that CHUK is a tertiary care and referral hospital, treating many patients who have already received first-line (access) antimicrobials prior to transfer. This finding is comparable to findings from studies conducted in tertiary settings in Northern and Southern Europe and other African countries [42,43,44,45]. Surprisingly, only 14.3% of patients subject to empirical treatment had any samples sent to the microbiology laboratory for culture and antibiotic susceptibility testing (AST). While this is comparable to findings from other PPS reports [42,43], the low number of samples sent to the laboratory suggests the insufficient use of microbiology tests by the treating physicians, possibly due to over-confidence of physicians in their empirical prescription practices [32] or to lowered expectations of receiving useful and timely AST results, due to the limited laboratory capacity. As the microbiology lab at CHUK often faces supply issues for blood culture bottles, antibiotic disks, and reagents required for culture and AST, some of this pessimism may be justified.

The samples sent for culture showed a worrying prevalence of multi-drug-resistant organisms, such as ESBL producing 3GCS-resistant Enterobacteriaceae, CPE, and carbapenem-resistant non-fermenter organisms. However, because of the low number of samples sent, no robust conclusions on the epidemiology of infectious pathogens at CHUK can be drawn from the current study.

The antimicrobial prescribing quality was suboptimal, as suspected, due to the absence of guidelines. This was illustrated by the low to very low levels of oral administration and the infrequent documentation of a stop or review date in the patient files. However, the reason for antimicrobial prescription was noted in the patient files in more than 90% of the cases. This is in line with several other African countries, which have performed a PPS [39,45]. The potential drivers are diverse and need further exploration, but they may include poor documentation by healthcare providers, a lower availability of drugs for oral use, a limited awareness of international guidelines, inadequate local data and the capacity to create local guidelines, and a limited awareness of the health care providers on the advantages of using oral antibiotics. 

Our survey has a couple of limitations. First, the PPS was conducted on different days depending upon the ward surveyed, due to logistic difficulties created by the limited study personnel available. However, each ward was entirely surveyed on a single day as per the protocol. Second, the PPS provides a snapshot of antimicrobial prescription data at a single time point, which cannot not capture changes in the hospital’s antimicrobial prescription practices over time. Therefore, additional measurements and interventions will be performed as part of a quality improvement cycle, such as auditing SP practices in order to confirm the findings of the PPS.

This is the first PPS conducted at the University Teaching Hospital of Kigali, implemented by healthcare providers who have been trained using the G-PPS tool. This study provides a baseline data for CHUK and serves as reference for other tertiary hospitals in Rwanda and regional LMIC settings. Furthermore, the data from this PPS will inform the design of multiple different interventions aimed at improving antimicrobial stewardship at CHUK. Such AMS programs have improved antimicrobials prescription practices in other settings [46,47]. 

## 4. Materials and Methods

This is a descriptive study that was conducted at CHUK, a tertiary-care and teaching hospital, located in Nyarugenge district, Rwanda. CHUK has a bed capacity of 519 beds for hospitalization and an average occupancy level of 81%. The PPS was conducted as part of the Antimicrobial Stewardship Program (ASP), following the protocol of the Global PPS of Antimicrobial Consumption and Resistance (Global-PPS) project (https://www.global-pps.com/) from 27 to 30 March 2023. The Global-PPS data collection was performed using the basic PPS protocol to survey inpatient antimicrobial use and the additional healthcare-associated infections (HAI) PPS module (Global-PPS protocol version 2023) (Ref: Appendix A).

Prior to conducting the PPS, the multidisciplinary teams consisting of nurses, physicians, pharmacists, microbiologists, and quality assurance personnel were trained on the usage of the G-PPS tool, and a pilot survey was conducted. The ethical approval to conduct the PPS was obtained from the CHUK division of Research, Education and Training. All data were collected from the patient files and included all inpatients occupying a bed at 8:00 AM on the day of the PPS. In addition to the data on the use of antimicrobials, we also collected patient demographics and medical information including the healthcare unit of admission. The collected data for all patients on antimicrobials at the time of the study included age, sex, biomarker of inflammation such as white blood cell count, C-reactive protein (CRP), and Procalcitonin (PCT), and information on the type and results of culture(s) of body fluids and/or tissue. Additionally, we collected information on the name of the prescribed antimicrobial, the start date, the dosage, and the administration route, the diagnosis and type of indication (healthcare-associated versus community-acquired infection, surgical or medical prophylaxis). Furthermore, we collected information about the documentation of a stop/review date of antimicrobial treatment and the reason for prescribing in the patient notes, the presence/absence of, as well as compliance to, (local) guidelines, and the number of missed doses. If the antimicrobial treatment was targeted based on microbiological results, the type of micro-organism(s) and their resistance profile(s) were noted. For the HAI module, additional data were collected for all patients on antimicrobials, including the presence of invasive devices and underlying morbidities.

Data analysis: The data were entered into the web-based application for Global-PPS data entry, an in-house built application for data entry, validation, and reporting, developed by the University of Antwerp. The data were extracted in Microsoft Excel (2021) and analyzed descriptively, and the results were expressed as frequencies and percentages. The antibiotics were categorized as access, watch, reserve, or not recommended following the 2023 WHO AWaRE classification “https://www.who.int/publications/i/item/WHO-MHP-HPS-EML-2023.04 (accessed on 24 January 2024)”. The HAI prevalence was calculated by dividing the number of patients receiving at least one antimicrobial for an HAI by the total number of admitted patients at the time of the survey.

## 5. Conclusions

This survey highlights different gaps that are associated with the irrational prescription and usage of antibiotics, and it calls for the urgent need to establish and implement a local and national antimicrobial stewardship program to optimize the usage of antibiotics and further explore the drivers of the prescribing behaviors.

## Figures and Tables

**Figure 1 antibiotics-13-01032-f001:**
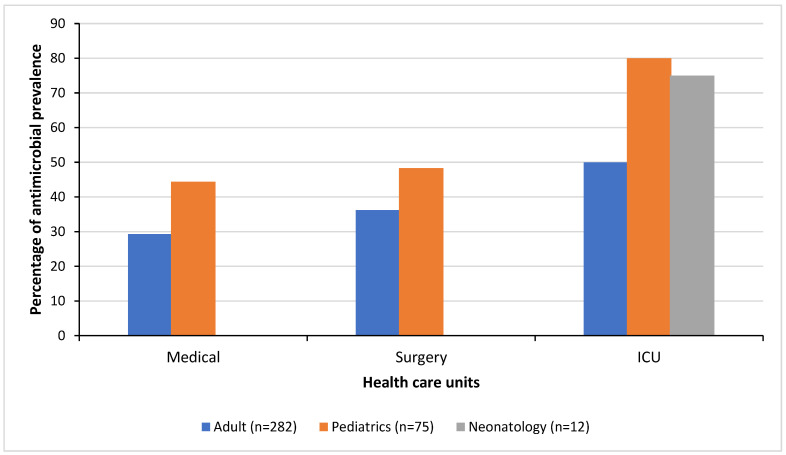
Antimicrobial prevalence per population groups and healthcare units.

**Figure 2 antibiotics-13-01032-f002:**
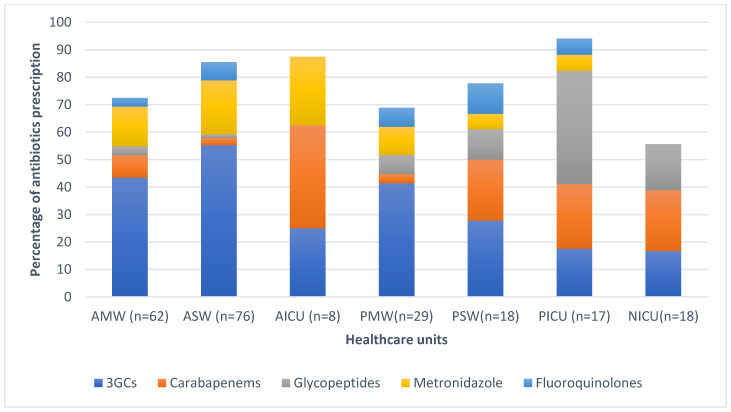
Distribution of the five most prescribed antimicrobials at healthcare unit level. Healthcare units include AMW: adult medical ward; ASW: adult surgical ward; AICU: adult intensive care unit; PMW: pediatric medical ward; PSW: pediatric surgical ward; PICU: pediatric intensive care unit; NICU: neonatology intensive care unit.

**Figure 3 antibiotics-13-01032-f003:**
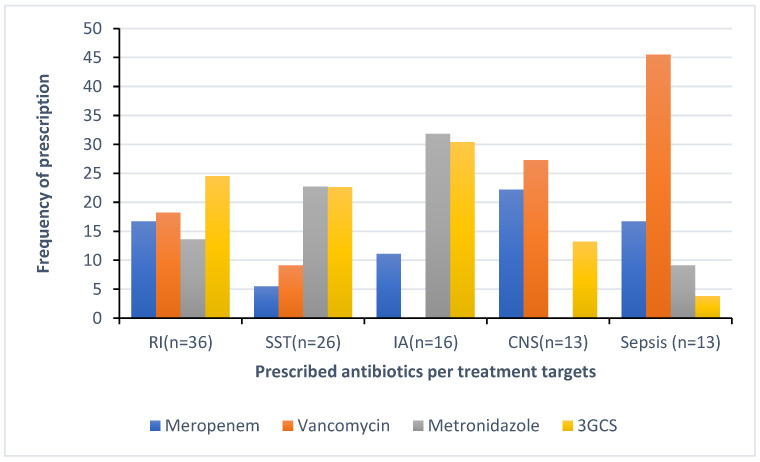
Distribution of most prescribed antibacterials in the top five diagnosis (TB excluded). The treatment targets include RI: respiratory infections including pneumonia and bronchitis; SST: skin and soft tissue infections; IA: intra-abdominal infections; CNS: central nervous system infections.

**Table 1 antibiotics-13-01032-t001:** Quality indicators of antimicrobial prescription (N= 259).

Indicators	Pediatric (*n* = 90)	Adult (*n* = 169)	Overall (*n* = 259)
Parenteral administration route	83 (92.2%)	122 (72.2%)	205 (79.1%)
Reason in notes	88 (97.8%)	151 (89.3%)	239 (92.3%)
Treatment based on biomarkers	33 (36.7%)	72 (42.6%)	105 (40.5%)
Sample for culture sent to the lab	13 (14.4%)	50 (29.6%)	63 (24.3%)
Targeted therapy	8 (8.9%)	31 (18.3%)	39 (15%)
Stop/review date documented	3 (3.3%)	24 (14.2%)	27 (10.4%)

## Data Availability

The supporting data of this manuscript can be made available on request from the corresponding author.

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
