# Peer review of "Prescription Practices and Usage of Antimicrobials in a Tertiary Teaching Hospital in Rwanda: A Call for Antimicrobial Stewardship"

_antibiotics, 2024, doi:10.3390/antibiotics13111032_

Round 1

Reviewer 1 Report

Comments and Suggestions for Authors

Drs Igizeneza and colleagues provide a point prevalence study on the use and prescribing patterns in tertiary hospital n Rwanda.

The authors included several important aspects in their assessment and include important data about mode of antibiotic administration, resistance patterns and use of diagnostic modalities. 

The introduction is succinct and puts the audit into a broader context and explains the framework delivered by the WHO. 

The results are presented in a logical and clear manner. The results are also discussed in a logical sequence, with potential factors identified to explain the results in a local and broader context. The authors discuss the limitations of the study as well. 

The authors could discuss the results of the current audit with regards to possible quality-improvement projects originating from this study. 

The list of references is appropriate. 

formal aspects:

Some abbreviations, although considered common knowledge (CRP, PCT,AST) should also be explained.

L142: SSI accounting for .. 

L202: Other ...factors 

Comments on the Quality of English Language

The language and grammar are of high standard. 

Author Response

Comment 1: The authors could discuss the results of the current audit with regards to possible quality-improvement projects originating from this study. 

Response 1: Thank you for pointing this out. From this PPS different interventions are being initiated. So far three improvement projects have been initiated: (1) Surgical Prophylaxis (SP) audit of the quality indicators; (2) Local Guideline development on SP and lastly (3) SP prescribing behavior diagnosis. Unfortunately, all those projects are still ongoing and not yet completed to have data that can be included in this current manuscript. Therefore, it will not be possible to discuss them in this manuscript. Thank you for your understanding.

Comment 2: Some abbreviations, although considered common knowledge (CRP, PCT,AST) should also be explained.

Response 2: Agree. We have therefore included the full of CRP and PCT on Page 8, paragraph 4 and line number 315-316;

AST was explained on page 7, paragraph 4, line 260;

SSI explained on page 5, paragraph 2 and line 158.

Thank you very much for taking your time to review this work.  Your invaluable comments are highly appreciated.

Reviewer 2 Report

Comments and Suggestions for Authors The work, all in all, is interesting. It would be appropriate to report the entire period of data enrollment, not just the beginning.The results are described and the data analyzed adequately, also underlining the weak points of the work due to the social background of the place of origin.The conclusions are clear. The references are adequate.it is advisable to insert, as previously stated, the period of enlistment.

Author Response

Comment 1: The work, all in all, is interesting. It would be appropriate to report the entire period of data enrollment, not just the beginning. The results are described and the data analyzed adequately, also underlining the weak points of the work due to the social background of the place of origin. The conclusions are clear. The references are adequate.it is advisable to insert, as previously stated, the period of enlistment.

Response 1: Thank you for pointing this out. We have included the entire period of enrollment was for 4 days from 27th-30th March 2023 as highlighted on page 8, paragraph 3, Line 303.

Thank you very much for taking your time to review this work.  Your invaluable comments are highly appreciated.

Reviewer 3 Report

Comments and Suggestions for Authors

Prescription practices and usage of antimicrobials in a tertiary  teaching hospital in Rwanda: A call for antimicrobial steward- ship

Dear author and editor,

The article talked about prescribing patterns of antibiotics in tertiary teaching hospital in Rwanda.

The article could be published after a minor revision.

I have some comments on it:

·        The discussion about antibiotic prescribing have to be written with more details and more citation.

·        the author should mention about antimicrobial steward-ship in the introduction and in the discussion.

·        The author should shed the light about the antimicrobial resistance problem in Rwanda and its spread in the world, with percentages about resistance cases.

Thank you very much, best regards.

Author Response

Comment 1: The discussion about antibiotic prescribing has to be written with more details and more citation.

Response 1:  Thank you for pointing this out. We agree with this comment, we therefore included a paragraph on antibiotics prescription practice at CHUK in the discussion section on page 6, paragraph 2, line 193-199.

Comment 2:  The author should mention about antimicrobial steward-ship in the introduction and in the discussion.

Response 2: Thank you for this comment. We agree with you. We therefore included information on antimicrobial stewardship in the Introduction part on page 2, paragraph 2, line 58-61. In the discussion part, please refer to page 8, paragraph 2, line 287-291.

Comment 3: The author should shed the light about the antimicrobial resistance problem in Rwanda and its spread in the world, with percentages about resistance cases.

Response 3: Thank you for pointing this out. We have added information on the raised comment accordingly. About Global antimicrobial resistance, we have included some data of the report from the Global Antimicrobial Resistance and Use Surveillance System (GLASS), 2022, Page 2, paragraph 1, Line 51-55.

AMR in Rwanda: Unfortunately, in Rwanda, no robust surveillance has been conducted so far to get the real figures of the antimicrobial resistance at country level. However, different studies have demonstrated the problem of resistance in Rwanda in referral hospitals. Therefore, we have added more literatures in that regards as found on page 2, paragraph 5, line 75-81.

Thank you very much for taking your time to review this work.  Your invaluable comments are highly appreciated.